# Injecting Prior Knowledge into Image Caption Generation

**Abstract.** Automatically generating natural language descriptions from an image is a challenging problem in artificial intelligence that requires a good understanding of the visual and textual signals and the correlations between them. The state-of-the-art methods in image captioning struggles to approach human level performance, especially when data is limited. In this paper, we propose to improve the performance of the state-of-the-art image captioning models by incorporating two sources of prior knowledge: (i) a conditional latent topic attention, that uses a set of latent variables (topics) as an anchor to generate highly probable words and, (ii) a regularization technique that exploits the inductive biases in syntactic and semantic structure of captions and improves the generalization of image captioning models. Our experiments validate that our method produces more human interpretable captions and also leads to significant improvements on the MSCOCO dataset in both the full and low data regimes.

## 1 Introduction

In recent years there has been a growing interest to develop end-to-end learning algorithms in computer vision tasks. Despite the success in many problems such as image classification [17] and person recognition [21], the state-of-the-art methods struggle to reach human-level performance in solving more challenging tasks such as image captioning within limited time and data which involves understanding the visual scenes and describing them in a natural language. This is in contrast to humans who are effortlessly successful in understanding the scenes which they have never seen before and communicating them in a language. It is likely that this efficiency is due to the strong prior knowledge of structure in the visual world and language [11].

Motivated by this observation, in this paper we ask "How can such prior knowledge be represented and utilized to learn better image captioning models with deep neural networks?". To this end, we look at the state-of-the-art encoder-decoder image captioning methods [39, 41, 3] where a Convolutional Neural Network (CNN) encoder extracts an embedding from the image, a Recurrent Neural Network (RNN) decoder generates the text based on the embedding. This framework typically contains two *dynamic* mechanisms to model the sequential output: i) an attention module [4, 41] that identifies the relevant parts of the image embedding based on the previous word and visual features and ii) the RNN decoder

Fig. 1: Our Final Model with Conditional Latent Topic Attention (CLTA) and Sentence Prior (Sentence Auto-Encoder (SAE) regularizer) both rely on prior knowledge to find relevant words and generate non-template like and generalized captions compared to the same Baseline caption for both images - *A man hitting a tennis ball with a racket.*

that predicts the next words based on the its previous state and attended visual features. While these two components are very powerful to model complex relations between the visual and language cues, we hypothesize that they are also capable of and at the same time prone to overfitting to wrong correlations, thus leading to poor generalization performance when the data is limited. Hence, we propose to regulate these modules with two sources of prior knowledge.

First, we propose an attention mechanism that accurately attends to relevant image regions and better cope with complex associations between words and image regions. For instance, in the example of a "man playing tennis", the input visual attention encoder might only look at the local features (*tennis ball*) leaving out the global visual information (*tennis court*). Hence, it generates a trivial caption as "A man is hitting a tennis ball", which is not the full description of the image in context (as shown in fig. 1). We solve this ambiguity by incorporating prior knowledge of latent topics [7], which are known to identify semantically meaningful topics [8], into our attention module. In particular we introduce a Conditional Latent Topic Attention (CLTA) module that models relationship between a word and image regions through a latent shared space *i.e.* latent topics to find salient regions in an image. *Tennis ball* steers the model to associate this word with the latent topic, "tennis", which further is responsible for localizing *tennis court* in the image. If a region-word pair has a higher probability with respect to a latent topic and if the same topic has a higher probability with respect to some other regions, then it is also a salient region and will be highly weighted. Therefore, we compute two sets of probabilities conditioned on the current word of the captioning model. We use conditional-marginalized probability where marginalization is done over latent topics to find salient image regions to generate the next word. Our CLTA is modeled as a neural network where marginalized probability is used to weight the image region features to obtain a context vector that is passed to a image captioning decoder to generate the next word.

Second, the complexity in the structure of natural language makes it harder to generate fluent sentences while preserving a higher amount of encoded infor-

mation (high Bleu-4 scores). Although current image captioning models are able to model this linguistic structure, the generated captions follow a more template-like form, for instance, "A man hitting a tennis ball with a racket." As shown in fig. 1, visually similar images have template-like captions from the baseline model. Inspired from sequence-to-sequence (seq2seq) machine translation [35, 28, 40, 16], we introduce a new regularization technique for captioning models coined SAE Regularizer. In particular, we design and train an additional seq2seq sentence auto-encoder model ("SAE") that first reads in a whole sentence as input, generates a fixed dimensional vector, then the vector is further used to reconstruct the input sentence. Human languages are highly structured and follows immense amount of regularity. Certain words are more likely to co-appear and certain word patterns can be observed more often. Our SAE is trained to learn the structure of the input (sentence) space in an offline manner by exploiting the regularity of the sentence space. The continuous latent space learned by SAE blends together both the syntactic and semantic information from the input sentence space and generates high quality sentences during the reconstruction via the SAE decoder. This suggests that the continuous latent space of SAE contains sufficient information regarding the syntactic and semantic structure of input sentences. Specifically, we use SAE-Dec as an auxiliary decoder branch (see fig. 3). Adding this regularizer forces the representation from the image encoder and language decoder to be more representative of the visual content and less likely to overfit. SAE-Dec is employed along with the original image captioning decoder ("IC-Dec") to output the target sentence during training, however, we do not use SAE regularizer at test time reducing additional computations.

Both of the proposed improvements also help to overcome the problem of training on large image-caption paired data [26, 27] by incorporating prior knowledge which is learned from unstructured data in the form of latent topics and SAE. These priors – also known as "inductive biases" – help the models make inferences that go beyond the observed training data. Through an extensive set of experiments, we demonstrate that our proposed CLTA module and SAE-Dec regularizer improves the image captioning performance both in the limited data and full data training regimes on the MSCOCO dataset [26].

## 2    Related Work

Here, we first discuss related attention mechanisms and then the use of knowledge transfer in image captioning models.

**Attention mechanisms in image captioning.**  The pioneering work in neural machine translation [4, 29, 9] has shown that attention in encoder-decoder architectures can significantly boost the performance in sequential generation tasks. Visual attention is one of the biggest contributor in image captioning [15, 41, 3, 19]. Soft attention and hard attention variants for image captioning were introduced in [41]. Bottom-Up and Top-Down self attention is effectively used in [3]. Attention on attention is used in recent work [19]. Interestingly, they use attention at both encoder and the decoder step of the captioning process. Our

proposed attention significantly differs in comparison to these attention mechanisms. First, the traditional attention methods, soft-attention [4] and scaled dot product attention [36] aims to find features or regions in an image that highly correlates with a word representation [3, 4, 34]. In contrast, our *conditional-latent topic attention* uses latent variables *i.e.*topics as anchors to find relationship between word representations and image regions (features). Some image regions and word representations may project to the same set of latent topics more than the others and therefore more likely to co-occur. Our method learns to model these relationships between word-representations and image region features using our latent space. We allow competition among regions and latent topics to compute two sets of probabilities to find salient regions. This competing strategy and our latent topics guided by pre-trained LDA topics [7] allow us to better model relationships between visual features and word representations. Hence, the neural structure and our attention mechanism is quite different from all prior work [41, 3, 19, 4].

**Knowledge transfer in image captioning.** It is well known that language consists of semantic and syntactic biases [5, 30]. We exploit these biases by first training a recurrent caption auto-encoder to capture this useful information using [35]. Our captioning auto-encoder is trained to reconstruct the input sentence and hence, this decoder encapsulates the structural, syntactic and semantic information of input captions. During captioning process we regularize the captioning RNN with this pretrained caption-decoder to exploit biases in the language domain and transfer them to the visual-language domain. To the best of our knowledge, no prior work has attempted such knowledge transfer in image captioning. Zhou *et al.*[46] encode external knowledge in the form of knowledge graphs using Concept-Net [27] to improve image captioning. The closest to ours is the work of [42] where they propose to generate scene graphs from both sentences and images and then encode the scene graphs to a common dictionary before decoding them back to sentences. However, generation of scene graphs from images itself is an extremely challenging task. Finally, we propose to transfer syntactic and semantic information as a regularization technique during the image captioning process as an auxiliary loss. Our experiments suggest that this leads to considerable improvements, specially in more structured measures such as CIDEr [37].

## 3   Method

In this section, we first review image captioning with attention, introduce our CLTA mechanism, and then our sentence auto-encoder (SAE) regularizer.

### 3.1   Image Captioning with Attention

Image captioning models are based on encoder-decoder architecture [41] that use a CNN as image encoder and a Long Short-Term Memory (LSTM) [18] as the decoder – see Fig.1.

The encoder takes an image as input and extracts a feature set $v = \{\boldsymbol{v}_1, \ldots, \boldsymbol{v}_R\}$ corresponding to $R$ regions of the image, where $\boldsymbol{v}_i \in \mathbb{R}^D$ is the $D$-dimensional feature vector for the $i^{th}$ region. The decoder outputs a caption $y$ by generating one word at each time step. At time step $t$, the feature set $v$ is combined into a single vector $\boldsymbol{v}_a^t$ by taking weighted sum as follows:

$$\boldsymbol{v}_a^t = \sum_{i=1}^{R} \alpha_i^t \boldsymbol{v}_i \qquad (1)$$

where $\alpha_i^t$ is the CLTA weight for region $i$ at time $t$, that is explained in the next section. The decoder LSTM $\phi$ then takes a concatenated vector $[\boldsymbol{v}_a^t | \boldsymbol{y}_{t-1}]$ and the previous hidden state $\mathbf{h_{t-1}}$ as input and generates the next hidden state $\mathbf{h_t}$:

$$\mathbf{h_t} = \phi([\boldsymbol{v}_a^t | E\boldsymbol{y}_{t-1}], \mathbf{h_{t-1}}, \Theta_\phi) \qquad (2)$$

where, $|$ denotes concatenation, $\boldsymbol{y}_{t-1} \in \mathbb{R}^K$ is the one-hot vector of the word generated at time $t-1$, $K$ is the vocabulary size, $\boldsymbol{h}^t \in \mathbb{R}^n$ is the hidden state of the LSTM at time $t$, $n$ is the LSTM dimensionality, and $\Theta_\phi$ are trainable parameters of the LSTM. Finally, the decoder predicts the output word by applying a linear mapping $\psi$ on the hidden state and $\boldsymbol{v}_a^t$ as follows:

$$\boldsymbol{y}_t = \psi([\mathbf{h_t} | \boldsymbol{v}_a^t], \Theta_\psi) \qquad (3)$$

where $\Theta_\psi$ are trainable parameters. Our LSTM implementation closely follows the formulation in [45]. The word embedding matrix $E \in \mathbb{R}^{m \times K}$ is trained to translate one-hot vectors to word embeddings as in [41], where $m$ is the word embedding dimension. In the next section, we describe our proposed CLTA mechanism.

### 3.2   CLTA: Conditional Latent Topic Attention

At time step $t$, our CLTA module takes the previous LSTM hidden state $(\boldsymbol{h}^{t-1})$ and image features to output the attention weights $\alpha^t$. Specifically, we use a set of latent topics to model the associations between textual $(\boldsymbol{h}^{t-1})$ and visual features $(\boldsymbol{v})$ to compute the attention weights. The attention weight for region $i$ is obtained by taking the conditional-marginalization over the latent topic $l$ as follows:

$$\alpha_i^t = P(\text{region} = i | h^{t-1}, \boldsymbol{v}) = \sum_{l=1}^{C} P(\text{region} = i | h^{t-1}, \boldsymbol{v}, l) P(l | h^{t-1}, \boldsymbol{v}_i) \qquad (4)$$

where $l$ is a topic variable in the $C$-dimensional latent space. To compute $P(l | h^{t-1}, \boldsymbol{v}_i)$, we first project both textual and visual features to a common $C$-dimensional shared latent space, and obtain the associations by summing the projected features as follows:

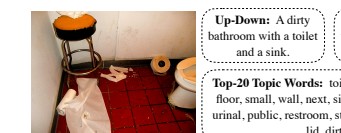
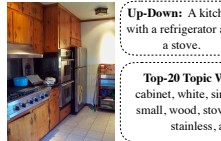

Fig. 2: Image-Caption pairs generated from our CLTA module with 128 dimensions and visualization of Top-20 words from the latent topics.

$$\boldsymbol{q}_i^t = W_{sc}\boldsymbol{v}_i + W_{hc}\boldsymbol{h}^{t-1} \tag{5}$$

where $W_{sc} \in \mathbb{R}^{C \times D}$ and $W_{hc} \in \mathbb{R}^{C \times n}$ are the trainable projection matrices for visual and textual features, respectively. Then the latent topic probability is given by:

$$P_L = P(l|\boldsymbol{h}^{t-1}, \boldsymbol{v}_i) = \frac{\exp(\boldsymbol{q}_{il}^t)}{\sum_{k=1}^{C} \exp(\boldsymbol{q}_{ik}^t)} \tag{6}$$

Afterwards, we compute the probability of a region given the textual, vision features and latent topic variable as follows:

$$\boldsymbol{r}_i^t = W_{sr}\boldsymbol{v}_i + W_{hr}\boldsymbol{h}^{t-1} \tag{7}$$

$$P(\text{region} = i|\boldsymbol{h}^{t-1}, v, l) = \frac{\exp(\boldsymbol{r}_{il}^t)}{\sum_{k=1}^{R} \exp(\boldsymbol{r}_{kl}^t)} \tag{8}$$

where $W_{sr} \in \mathbb{R}^{C \times D}$ and $W_{hr} \in \mathbb{R}^{C \times n}$ are the trainable projection matrices for visual and textual features, respectively.

The latent topic posterior in eq. (6) is pushed to the pre-trained LDA topic prior by adding a KL-divergence term to the image captioning objective. We apply Latent Dirichlet Allocation (LDA) [7] on the caption data. Then, each caption has an inferred topic distribution $Q_T$ from the LDA model which acts as a prior on the latent topic distribution, $P_L$. For doing this, we take the average of the C-dimensional latent topics at all time steps from $0, \ldots, t-1$ as:

$$P_{L_{avg}} = \frac{1}{t} \sum_{k=0}^{t-1} P(l|\boldsymbol{h}^k, \boldsymbol{v}_i) \tag{9}$$

Hence, the KL-divergence objective is defined as:

$$D_{KL}(P_{L_{avg}}||Q_T) = \sum_{c \in C} P_{L_{avg}}(c) \times log(\frac{P_{L_{avg}}(c)}{Q_T(c)}) \tag{10}$$

This learnt latent topic distribution captures the semantic relations between the visual and textual features in the form of visual topics, and therefore we also use this latent posterior, $P_L$ as a source of meaningful information during

generation of the next hidden state. The modified hidden state $\mathbf{h_t}$ in eq. (2) is now given by:

$$\mathbf{h_t} = \phi([\boldsymbol{v}_a^t|E\boldsymbol{y}_{t-1}|P_L], \mathbf{h_{t-1}}, \Theta_\phi) \quad (11)$$

We visualize the distribution of latent topics in Figure 2. While traditional "soft-max" attention exploit simple correlation among textual and visual information, we make use of latent topics to model associations between them.

### 3.3   SAE Regularizer

Encoder-decoder methods are widely used for translating one language to another [10, 35, 4]. When the input and target sentences are the same, these models function as auto-encoders by first encoding an entire sentence into a fixed-(low) dimensional vector in a latent space, and then reconstructing it. Autoencoders are commonly employed for unsupervised training in text classification [13] and machine translation [28].

In this paper, our SAE regularizer has two advantages: i) acts as a soft constraint on the image captioning model to regularize the syntactic and semantic space of the captions for better generalization and, ii) encourages the image captioning model to extract more context information for better modelling long-term memory. These two properties of the SAE regularizer generates semantically meaningful captions for an image with syntactic generalizations and prevents generation of naive and template-like captions.

Our SAE model uses network architecture of [35] with Gated Recurrent Units (GRU) [12]. Let us denote the parameter of the decoder GRU by $\Theta_{\mathrm{D}}$. A stochastic variation of the vanilla sentence auto-encoders is de-noising auto-encoders [38] which are trained to "de-noise" corrupted versions of their inputs. To inject such input noise, we drop each word in the input sentence with a probability of 50% to reduce the contribution of a single word on the semantics of a sentence. We train the SAE model in an offline stage on training set of the captioning dataset. After the SAE model is trained, we discard its encoder and integrate only its decoder to regularize the captioning model.

As depicted in Figure 3, the pretrained SAE decoder takes the last hidden state vector of captioning LSTM $\boldsymbol{h}$ as input and generates an extra caption (denoted as $y_{\mathrm{sae}}$) in addition to the output of the captioning model (denoted as $y_{\mathrm{lstm}}$). We use output of the SAE decoder only in train time to regulate the captioning model $\phi$ by implicitly transferring the previously learned latent structure with SAE decoder.

Our integrated model is optimized to generate two accurate captions (*i.e.* $y_{\mathrm{sae}}$ and $y_{\mathrm{lstm}}$) by minimizing a weighted average of two loss values:

$$\arg\min_{\Omega} \quad \lambda L(y^*, y_{\mathrm{lstm}}) + (1 - \lambda)L(y^*, y_{\mathrm{sae}}) \quad (12)$$

where $L$ is the cross-entropy loss computed for each caption, word by word against the ground truth caption $y^*$, $\lambda$ is the trade-off parameter, and $\Omega$ are

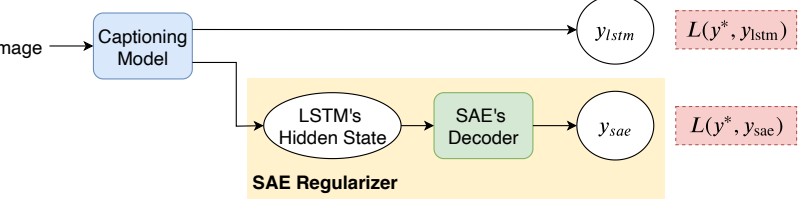

Fig. 3: Illustration of our proposed Sentence Auto-Encoder (SAE) regularizer with the image captioning decoder. The captioning model is trained by adding the SAE decoder as an auxiliary branch and thus acting as a regularizer.

the parameters of our model. We consider two scenarios that we use during our experimentation.

- First, we set the parameters of the SAE decoder $\Theta_D$ to be the weights of the pre-trained SAE decoder and freeze them while optimizing Equation (12) in terms of $\Omega = \{\Theta_\phi, \Theta_\psi, E\}$.
- Second, we initialize $\Theta_D$ with the weights of the pre-trained SAE decoder and fine-tune them along with the LSTM parameters, $i.e.\,\Omega = \{\Theta_\phi, \Theta_\psi, E, \Theta_D\}$.

As discussed in section 3.2, we also minimize the KL divergence in eq. (10) along with the final regularized objective in eq. (12) as:

$$\arg\min_{\Omega} \quad \lambda L(y^*, y_{\text{lstm}}) + (1 - \lambda)L(y^*, y_{\text{sae}}) + \gamma D_{KL}(P_{L_{avg}}||Q_T) \qquad (13)$$

where, $\gamma$ is the weight for the KL divergence loss.

*Discussion.* An alternative way of exploiting the information from the pre-trained SAE model is to bring the representations from the captioning decoder closer to the encodings of the SAE encoder by minimizing the Euclidean distance between the hidden state from the SAE encoder and the hidden state from the captioning decoder at each time-step. However, we found this setting is too restrictive on the learned hidden state of the LSTM.

## 4   Experiments

**Dataset.** Our models are evaluated on the standard MSCOCO 2014 image captioning dataset [26]. For fair comparisons, we use the same data splits for training, validation and testing as in [22] which have been used extensively in prior works. This split has 113,287 images for training, 5k images for validation and testing respectively with 5 captions for each image. We perform evaluation on all relevant metrics for generated sentence evaluation - CIDEr [37], Bleu [31], METEOR [14], ROUGE-L [25] and, SPICE [2].

**Implementation Details.** For training our image captioning model, we compute the image features based on the Bottom-Up architecture proposed by [3], where the model is trained using a Faster-RCNN model [32] on the Visual-Genome Dataset [24] with object and attribute information. These features are extracted from $R$ regions and each region feature has $D$ dimensions, where $R$ and $D$ is 36 and 2048 respectively as proposed in [3]. We use these $36 \times 2048$ image features in all our experiments.

## 4.1 Experimental Setup

*LDA Topic Models.* The LDA [7] model is learned in an offline manner to generate a $C$ dimensional topic distribution for each caption. Briefly, the LDA model treats the captions as word-documents and group these words to form $C$ topics (cluster of words), learns the word distribution for each topic ($C \times V$) where $V$ is the vocabulary size and also generates a topic distribution for each input caption, $Q_T$ where each $C^{th}$ dimension denotes the probability for that topic.

*Sentence Auto-Encoder.* The Sentence Auto-encoder is trained offline on the MSCOCO 2014 captioning dataset [26] with the same splits as discussed above. For the architecture, we have a single layer GRU for both the encoder and the decoder. The word embeddings are learned with the network using an embedding layer and the dimension of both the hidden state and the word embeddings is 1024. During training, the decoder is trained with teacher-forcing [6] with a probability of 0.5. For inference, the decoder decodes till it reaches the end of caption token. The learning rate for this network is 2e-3 and it is trained using the ADAM [23] optimizer.

*Image Captioning Decoder with SAE Regularizer.* The architecture of our image captioning decoder is same as the Up-Down model [3] with their "soft-attention" replaced by our CLTA module and trained with the SAE regularizer. We also retrain the AoANet model proposed by Huang *et al.*[19] by incorporating our CLTA module and the SAE regularizer. In the results section, we show improvements over the Up-Down and AoANet models using our proposed approaches. Note, the parameters for training Up-Down and AoANet baselines are same as the original setting. While training the captioning models together with the SAE-decoder, we jointly learn an affine embedding layer (dimension 1024) by combining the embeddings from the image captioning decoder and the SAE-decoder. During inference, we use beam search to generate captions from the captioning decoder using a beam size of 5 for Up-Down and a beam-size of 2 for AoANet. For training the overall objective function as given in Equation 13, the value of $\lambda$ is initialized by 0.7 and increased by a rate of 1.1 every 5 epochs until it reaches a value of 0.9 and $\gamma$ is fixed to 0.1. We use the ADAM optimizer with a learning rate of 2e-4. Our code is implemented using PyTorch [1] and will be made publicly available.

## 5    Results and Analysis

First, we study the caption reconstruction performance of vanilla and denoising SAE, then report our model's image captioning performance on MS-COCO dataset with full and limited data, investigate multiple design decisions and analyze our results qualitatively.

### 5.1    Sentence Auto-Encoder Results

An ideal SAE must learn mapping its input to a fixed low dimensional space such that a whole sentence can be summarized and reconstructed accurately. To this end, we experiment with two SAEs, Vanilla-SAE and Denoising-SAE and report their reconstruction performances in terms of Bleu4 and cross-entropy (CE) loss in fig.4. The vanilla model, when the inputs words are not corrupted, outperforms the denoising one in both metrics. This is expected as the denoising model is only trained with corrupted input sequences. The loss for both the Vanilla and Denoising SAE start from a relatively high value of approximately 0.8 and 0.4 respectively, and converge to a significantly low error of 0.1 and 0.2. For a better analysis, we also compute the Bleu-4 metrics on our decoded caption against the 5 ground-truth captions. As reported in fig.1, both models obtain significantly high Bleu-4 scores. This indicates that an entire caption can be compressed in a low dimensional vector (1024) and can be successfully reconstructed.

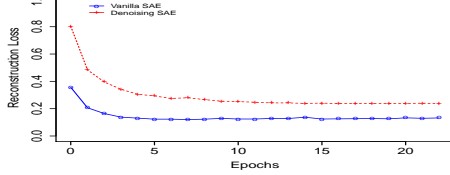

Fig. 4: Error Curve for the Sentence Auto-Encoder on the Karpathy test split. The error starts increasing approximately after 20 epochs.

| Models | Bleu-4 ↑ | CE-Loss ↓ |
|---|---|---|
| Vanilla SAE | **96.33** | **0.12** |
| Denoising SAE | 89.79 | 0.23 |

Table 1: Bleu-4 Evaluation and Reconstruction Cross-Entropy Loss for the Sentence Auto-Encoder on the Karpathy test split of MSCOCO 2014 caption dataset [26].

### 5.2    Image Captioning Results

Here we incorporate the proposed CLTA and SAE regularizer to recent image-captioning models including Up-Down [3] and AoANet [19] and report their performance on MS-COCO dataset in multiple metrics (see Table 2). The tables report the original results of these methods from their publications in the top block and the rows in cyan show relative improvement of our models when compared to the baselines.

The baseline models are trained for two settings - 1)Up-Down†, is the model re-trained on the architecture of Anderson *et al.*[3] and, 2) AoANet†, is the

| Models | cross-entropy loss | | | | | | cider optimization | | | | | |
|---|---|---|---|---|---|---|---|---|---|---|---|---|
| | B-1 | B-4 | M | R | C | S | B-1 | B-4 | M | R | C | S |
| LSTM-A [44] | 75.4 | 35.2 | 26.9 | 55.8 | 108.8 | 20.0 | 78.6 | 35.5 | 27.3 | 56.8 | 118.3 | 20.8 |
| RFNet [20] | 76.4 | 35.8 | 27.4 | 56.8 | 112.5 | 20.5 | 79.1 | 36.5 | 27.7 | 57.3 | 121.9 | 21.2 |
| Up-Down [3] | 77.2 | 36.2 | 27.0 | 56.4 | 113.5 | 20.3 | 79.8 | 36.3 | 27.7 | 56.9 | 120.1 | 21.4 |
| GCN-LSTM [43] | 77.3 | 36.8 | 27.9 | 57.0 | 116.3 | 20.9 | 80.5 | 38.2 | 28.5 | 58.3 | 127.6 | 22.0 |
| AoANet [19] | 77.4 | 37.2 | 28.4 | 57.5 | 119.8 | 21.3 | 80.2 | 38.9 | 29.2 | 58.8 | 129.8 | 22.4 |
| Up-Down$^\dagger$ | 75.9 | 36.0 | 27.3 | 56.1 | 113.3 | 20.1 | 79.2 | 36.3 | 27.7 | 57.3 | 120.8 | 21.2 |
| Up-Down$^\dagger$ + CLTA + SAE-Reg | **76.7** | **37.1** | **28.1** | **57.1** | **116.2** | **21.0** | **80.2** | **37.4** | **28.4** | **58.1** | **127.4** | **22.0** |
| Relative Improvement | +0.8 | +1.1 | +0.8 | +1.0 | +2.9 | +0.9 | +1.0 | +1.1 | +0.7 | +0.8 | +6.6 | +0.8 |
| AoANet$^*$ | 77.3 | 36.9 | **28.5** | 57.3 | 118.4 | 21.6 | 80.5 | 39.1 | 29.0 | 58.9 | 128.9 | 22.7 |
| AoANet$^\dagger$ + CLTA + SAE-Reg | **78.1** | **37.9** | 28.4 | **57.5** | **119.9** | **21.7** | **80.8** | **39.3** | **29.1** | **59.1** | **130.1** | **22.9** |
| Relative Improvement | +0.8 | +1.0 | -0.1 | +0.2 | +1.5 | +0.1 | +0.3 | +0.2 | +0.1 | +0.2 | +1.2 | +0.2 |

Table 2: Image captioning performance on the "Karpathy" test split of the MSCOCO 2014 caption dataset [26] from other state-of-the-art methods and our models. Our Conditional Latent Topic Attention with the SAE regularizer significantly improves across all the metrics using both *cross-entropy loss* and *cider optimization*. † denotes our trained models and * indicates the results obtained from the publicly available pre-trained model.

Attention-on-Attention model re-trained as in Huang *et al.*[19]. Note that for both Up-Down and AoANet, we use the original source code to train them in our own hardware. We replace the "soft-attention" module in our Up-Down baseline by CLTA directly. The AoANet model is based on the powerful Transformer [36] architecture with the multi-head dot attention in both encoder and decoder. For AoANet, we replace the dot attention in the decoder of AoANet at each head by the CLTA which results in multi-head CLTA. The SAE-decoder is added as a regularizer on top of these models as also discussed in section 4.1. As discussed later in section 5.5, we train all our models with 128 dimensions for the CLTA and with the Denoising SAE decoder (initialized with $\boldsymbol{h}^{last}$).

We evaluate our models with the cross-entropy loss training and also by using the CIDEr score oprimization [33] after the cross-entropy pre-training stage (table 2). For the cross-entropy one, our combined approach consistently improves over the baseline performances across all metrics. It is clear from the results that improvements in CIDEr and Bleu-4 are quite significant which shows that our approach generates more human-like and accurate sentences. It is interesting to note that AoANet with CLTA and SAE-regularizer also gives consistent improvements despite having a strong transformer language model. We show in section 5.4 the differences between our captions and the captions generated from Up-Down and AoANet. Our method is modular and improves on state-of-the-art models despite the architectural differences. Moreover, the SAE decoder is discarded after training and hence it brings no additional computational load during test-time but with significant performance boost. For CIDEr optimization, our models based on Up-Down and AoANet also show significant improvements in all metrics for our proposed approach.

| Models | 50% data | | 75% data | | 100% data | |
|---|---|---|---|---|---|---|
| | Bleu-4 | CIDEr | Bleu-4 | CIDEr | Bleu-4 | CIDEr |
| Up-Down | 35.4 | 112.0 | 35.8 | 112.7 | 36.0 | 113.3 |
| Up-Down+CLTA | 36.3 | 113.7 | 36.3 | 114.5 | 36.5 | 115.0 |
| Up-Down+CLTA+SAE-Reg | **36.6** | **114.8** | **36.8** | **115.6** | **37.1** | **116.2** |
| AoANet | 36.6 | 116.1 | 36.8 | 118.1 | 36.9 | 118.4 |
| AoANet+CLTA | 36.9 | 116.7 | 37.1 | 118.4 | 37.4 | 119.1 |
| AoANet+CLTA+SAE-Reg | **37.2** | **117.5** | **37.6** | **118.9** | **37.9** | **119.9** |

Table 3: Evaluation of our CLTA and SAE-Regularizer methods by training on a subset of the MSCOCO "Karpathy" Training split.

## 5.3 Learning to Caption with Less Data

Table 3 evaluates the performance of our proposed models for a subset of the training data, where $x\%$ is the percentage of the total data that is used for training. All these subsets of the training samples are chosen randomly. Our CLTA module is trained with 128 dimensions for the latent topics along with the Denoising SAE Regularizer initialized with the last hidden state of the LSTM (Up-Down+CLTA+SAE-Reg). Despite the number of training samples, our average improvement with CLTA and SAE-Regularizer is around 1% in Bleu-4 and 2.9% in CIDEr for the Up-Down model and 0.8% in Bleu-4 and 1.2% in CIDEr for the AoANet model. The significant improvements in Bleu-4 and CIDEr scores with only 50% and 75% of the data compared to the baseline validates our proposed methods as a form of rich prior.

## 5.4 Qualitative Results

In fig. 5, we show examples of images and captions generated by the baselines Up-Down and AoANet along with our proposed methods, CLTA and SAE-Regularizer. The baseline models have repetitive words and errors while generating captions (*in front of a mirror*, *a dog in the rear view mirror*). Our models corrects these mistakes by finding relevant words according to the context and putting them together in a human-like caption format (*a rear view mirror shows a dog* has the same meaning as *a rear view mirror shows a dog in the rear view mirror* which is efficiently corrected by our models by bringing in the correct meaning). From all the examples shown, we can see that our model overcomes the limitation of overfitting in current methods by completing a caption with more semantic and syntactic generalization (*e.g.*: *different flavoured donuts* and *several trains on the tracks*).

## 5.5 Ablation Study

**Conditional Latent Topic Attention (CLTA).** Table 4a depicts the results for the CLTA module that is described in section 3.2. Soft-attention is used as a

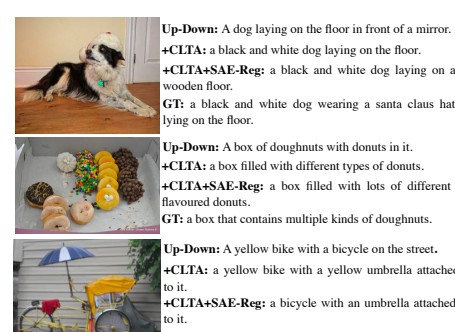

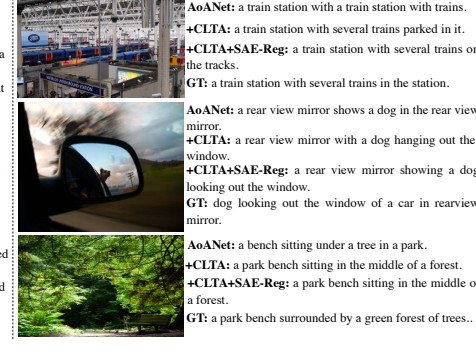

**Up-Down:** A dog laying on the floor in front of a mirror.
**+CLTA:** a black and white dog laying on the floor.
**+CLTA+SAE-Reg:** a black and white dog laying on a wooden floor.
**GT:** a black and white dog wearing a santa claus hat lying on the floor.

**Up-Down:** A box of doughnuts with donuts in it.
**+CLTA:** a box filled with different types of donuts.
**+CLTA+SAE-Reg:** a box filled with lots of different flavoured donuts.
**GT:** a box that contains multiple kinds of doughnuts.

**Up-Down:** A yellow bike with a bicycle on the street.
**+CLTA:** a yellow bike with a yellow umbrella attached to it.
**+CLTA+SAE-Reg:** a bicycle with an umbrella attached to it.
**GT:** a bicycle with an umbrella and a basket.

**AoANet:** a train station with a train station with trains.
**+CLTA:** a train station with several trains parked in it.
**+CLTA+SAE-Reg:** a train station with several trains on the tracks.
**GT:** a train station with several trains in the station.

**AoANet:** a rear view mirror shows a dog in the rear view mirror.
**+CLTA:** a rear view mirror with a dog hanging out the window.
**+CLTA+SAE-Reg:** a rear view mirror showing a dog looking out the window.
**GT:** dog looking out the window of a car in rearview mirror.

**AoANet:** a bench sitting under a tree in a park.
**+CLTA:** a park bench sitting in the middle of a forest.
**+CLTA+SAE-Reg:** a park bench sitting in the middle of a forest.
**GT:** a park bench surrounded by a green forest of trees..

Fig. 5: Example of generated captions from the baseline Up-Down, AoANet, our proposed CLTA and, our final models with both CLTA and SAE Regularizer.

baseline and corresponds to the attention mechanism in [41] which is the main attention module in Up-Down image captioning model by Anderson *et al.*[3]. We replace this attention with the CLTA and evaluate its performance for different number of latent dimensions, *i.e.* topics ($C$). The models trained with latent topic dimensions of 128, 256 and 512 all outperform the baseline significantly. The higher CIDEr and Bleu-4 scores for these latent topics show the model's capability to generate more descriptive and accurate human-like sentences.

As we increase the dimensions of latent topics from 128 to 512, we predict more relevant keywords as new topics learnt by the CLTA module with 512 dimensions are useful in encoding more information and hence generating meaningful captions.

| Models | Baseline | CLTA | | |
|---|---|---|---|---|
| | Soft-Attention | 128 | 256 | 512 |
| Bleu-4 | 36.0 | 36.5 | 36.6 | **36.7** |
| CIDEr | 113.3 | 115.0 | 115.2 | **115.3** |

(a) Evaluation scores for the Up-Down model with soft-attention and ablations of our CLTA module.

| Models | SAE-Decoder | $h$ | Bleu-4 | CIDEr |
|---|---|---|---|---|
| Baseline | No | - | 36.0 | 113.3 |
| CLTA-128 | Vanilla | First | 36.9 | 115.8 |
| | | Last | 36.8 | 115.3 |
| | Denoising | First | 36.8 | 116.1 |
| | | Last | 37.1 | **116.2** |
| CLTA-512 | Denoising | Last | **37.2** | 115.9 |

(b) Additional quantitative evaluation results from different settings of the SAE decoder when trained with image captioning decoder. $h$ denotes the hidden state.

Table 4: Ablative Analysis for different settings on our (a) CLTA module and, (b) SAE regularizer training.

**Image Captioning Decoder with SAE Regularizer.** Table 4b reports ablations for our full image captioning model (Up-Down with CLTA) and the SAE regularizer. As discussed in section 3.3, SAE decoder (parameters defined by

$\Theta_D$) is initialized with the hidden state of the image captioning decoder. During training, we test different settings of how the SAE decoder is trained with the image captioning decoder: (1) Vanilla vs Denoising SAE and, (2) $h^{\text{first}}$ vs $h^{\text{last}}$, whether the SAE decoder is initialized with the first or last hidden state of the LSTM decoder. For all the settings, we fine-tune the parameters of $\text{GRU}_{\text{D}}$ ($\Theta_D$) when trained with the image captioning model (the parameters are initialized with the weights of the pre-trained Vanilla or Denoising SAE decoder).

The results in Table 4b are reported on different combinations from the settings described above, with the CLTA having 128 and 512 dimensions in the image captioning decoder. Adding the auxiliary branch of SAE decoder significantly improves over the baseline model with CLTA and in the best setting, Denoising SAE with $h^{\text{last}}$ improves the CIDEr and Bleu-4 scores by 1.2 and 0.6 respectively. As the SAE decoder is trained for the task of reconstruction, fine-tuning it to the task of captioning improves the image captioning decoder.

Initializing the Vanilla SAE decoder with $h^{\text{last}}$ does not provide enough gradient during training and quickly converges to a lower error, hence this brings lower generalization capacity to the image captioning decoder. As $h^{\text{first}}$ is less representative of an entire caption compared to $h^{\text{last}}$, vanilla SAE with $h^{\text{first}}$ is more helpful to improve the captioning decoder training. On the other hand, the Denoising SAE being robust to noisy summary vectors provide enough training signal to improve the image captioning decoder when initialized with either $h^{\text{first}}$ or $h^{\text{last}}$ but slightly better performance with $h^{\text{last}}$ for Bleu-4 and CIDEr as it forces $h^{\text{last}}$ to have an accurate lower-dim representation for the SAE and hence better generalization. It is clear from the results in table 4b, that Denoising SAE with $h^{\text{last}}$ helps to generate accurate and generalizable captions. From our experiments, we found that CLTA with 128 topics and Denoising SAE (with $h^{\text{last}}$) has better performance than even it's counterpart with 512 topics. Hence, for all our experiments in section 5.2 and section 5.3 our topic dimension is 128 with Denoising SAE initialized with $h^{\text{last}}$.

## 6   Conclusion

In this paper, we have introduced two novel methods for image captioning that exploit prior knowledge and hence help to improve state-of-the-art models even when the data is limited. The first method exploits association between visual and textual features by learning latent topics via an LDA topic prior and obtains robust attention weights for each image region. The second one is an SAE regularizer that is pre-trained in an autoencoder framework to learn the structure of the captions and is plugged into the image captioning model to regulate its training. Using these modules, we obtain consistent improvements on two investigate models, bottom-up top-down and the AoANet image captioning model, indicating the usefulness of our two modules as a strong prior. In future work, we plan to further investigate potential use of label space structure learning for other challenging vision tasks with limited data and to improve generalization.

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
