# OpenReview forum: "Injecting Prior Knowledge into Image Caption Generation"
_thecvf.com/ECCV/2020/Workshop/VIPriors — VIPriors Oral_

### Official Review · AnonReviewer2 · 2020-07-21
**Powerful implementation of prior knowledge into image captioning models**

**Confidence:** 3
**Rating:** 9

**Review:**

[Summary] In 2-3 sentences, describe the key ideas, experiments, and their significance.

The authors propose two adding to prior knowledge based modules to image captioning models. One module uses prior knowledge on the association of keywords to image regions. The other module regularizes generated captions to be more realistic.

[Strengths] What are the strengths of the paper? Clearly explain why these aspects of the paper are valuable.

Simple but powerful ideas; clear methods; topical submission; excellent writing.

[Weaknesses] What are the weaknesses of the paper? Clearly explain why these aspects of the paper are weak.

By nature of the method experimental settings are complex;

[Overall rating] Paper rating: Strong accept

[Confidence] 4/5

[Detailed comments] Additional comments regarding the paper (e.g. typos or other possible improvements you would like to see for the camera-ready version of the paper, if any.)

- Grammar: lines 40, 189
- Formatting: line 221

---

### Official Review · AnonReviewer1 · 2020-07-28
**Injecting Prior Knowledge into Image Caption Generation**

**Confidence:** 4
**Rating:** 9

**Review:**

1. [Summary] In 2-3 sentences, describe the key ideas, experiments, and their significance.

 The paper tries to mitigate overfitting and generating easy captions by introducing prior knowledge from the dataset during training. To this end, authors propose to add visual-semantic relation prior knowledge by defining a series of Latent Topics, and semantic prior knowledge by training a Seq2seq module with the text. While the former is introduced in the training procedure as a self-attention with image region features, the latter is utilized to remove visual biased on semantic structures. Apart from increasing results of state-of-the-art approaches, they demonstrate that with their approach, image captioning models can rely on less data when training.

2. [Strengths] What are the strengths of the paper? Clearly explain why these aspects of the paper are valuable.

 -	The paper is easy to read. Ideas are easy to follow.
 -	It is very well motivated.
 -	Benefits of both modules (CLTA and SAE Regularizer) are clearly demonstrated in the experiments.
 -	The implementation is very well explained in detail.
 -	The benefits of adding prior knowledge (visual and semantic) is showed.
 -	Additionally, authors demonstrate the relevance of prior knowledge as it allows to train models with less data.

3. [Weaknesses] What are the weaknesses of the paper? Clearly explain why these aspects of the paper are weak.

 -	Although the improvement exists, in some situations it is marginal.

4. [Overall rating] Paper rating.

 9

5. [Justification of rating] Please explain how the strengths and weaknesses aforementioned were weighed in for the rating.

 Good paper. Well written, well motivated, simple method and positive results. On top of that, very much in line with the workshop.

6. [Detailed comments] Additional comments regarding the paper (e.g. typos or other possible improvements you would like to see for the camera-ready version of the paper, if any.)

---

### Decision · Program_Chairs · 2020-07-29

**Decision:**

Accept (Oral)

**Comment:**

It is our pleasure to inform you that your paper has been accepted to the oral track of the 1st Visual Inductive Priors for Data-Efficient Deep Learning Workshop.

Please note the following deadlines:
* August 11, 2020 - workshop material, including:
 * paper in PDF format;
 * pre-recorded video presentation;
 * slides of the presentation in PDF.
* September 15, 2020 - camera-ready paper

The reviews can be found on OpenReview. Please take these comments and suggestions into account when preparing the camera-ready version of your paper, which is due September 15, 2020. The camera-ready paper should be uploaded to OpenReview.

As part of the workshop, each paper for oral presentation must submit a pre-recorded 5 minute talk before August 11, 2020. You will receive more information on how to upload the material shortly. The requirements for the video are:
* Duration: maximum 5 minutes
* MP4 format
* File size max. 100 MB
* Has an inset with a video of the speaker
* 16:9 aspect ratio (strongly preferred)
* 1920x1080 resolution (strongly preferred, at least 720 height)

Our suggested software for pre-recording your presentation is Zoom. For more information, please refer to the following guides:
How to record with Zoom Guide: http://homepages.inf.ed.ac.uk/rbf/ECCV2020HowtoRecordusingZoom.pdf
How to Record with Zoom tutorial: https://www.youtube.com/watch?v=CR199W7HdC0
Please ensure that at least one of the authors of the paper is available to attend the workshop during the allotted times. Note that the workshop will take place in two sessions spread across time zones (details are to follow). We will send instructions on how to connect to the workshop as soon as possible. The schedule for all talks and papers will be posted soon at the workshop website: https://vipriors.github.io.

We look forward to seeing you at the workshop!